# Age, Sex, Body Mass Index, Diet and Menopause Related Metabolites in a Large Homogeneous Alpine Cohort

**DOI:** 10.3390/metabo12030205

**Published:** 2022-02-24

**Authors:** Vinicius Verri Hernandes, Nikola Dordevic, Essi Marjatta Hantikainen, Baldur Bragi Sigurdsson, Sigurður Vidir Smárason, Vanessa Garcia-Larsen, Martin Gögele, Giulia Caprioli, Ilaria Bozzolan, Peter P. Pramstaller, Johannes Rainer

**Affiliations:** 1Institute for Biomedicine (Affiliated to the University of Lübeck), Eurac Research, 39100 Bozen, Italy; vinicius.verri@eurac.edu (V.V.H.); nikola.dordevic@eurac.edu (N.D.); essimarjatta.hantikainen@eurac.edu (E.M.H.); baldurbs@landspitali.is (B.B.S.); smarason.sigurdur@gmail.com (S.V.S.); martin.goegele@eurac.edu (M.G.); giulia.caprioli@eurac.edu (G.C.); ilaria.bozzolan@eurac.edu (I.B.); peter.pramstaller@eurac.edu (P.P.P.); 2Department of Clinical Biochemistry, Landspitali—University Hospital, 108 Reykjavik, Iceland; 3BASF SE, 67063 Ludwigshafen, Germany; 4Program in Human Nutrition, Department of International Health, The Johns Hopkins Bloomberg School of Public Health, Baltimore, MD 21205, USA; vgla@jhu.edu

**Keywords:** metabolomics, menopause, gender differences, diet, aging, body mass index

## Abstract

Metabolomics in human serum samples provide a snapshot of the current metabolic state of an individuum. Metabolite concentrations are influenced by both genetic and environmental factors. Concentrations of certain metabolites can further depend on age, sex, menopause, and diet of study participants. A better understanding of these relationships is pivotal for the planning of metabolomics studies involving human subjects and interpretation of their results. We generated one of the largest single-site targeted metabolomics data sets consisting of 175 quantified metabolites in 6872 study participants. We identified metabolites significantly associated with age, sex, body mass index, diet, and menopausal status. While most of our results agree with previous large-scale studies, we also found novel associations including serotonin as a sex and BMI-related metabolite and sarcosine and C2 carnitine showing significantly higher concentrations in post-menopausal women. Finally, we observed strong associations between higher consumption of food items and certain metabolites, mostly phosphatidylcholines and lysophosphatidylcholines. Most, and the strongest, relationships were found for habitual meat intake while no significant relationships were found for most fruits, vegetables, and grain products. Summarizing, our results reconfirm findings from previous population-based studies on an independent cohort. Together, these findings will ultimately enable the consolidation of sets of metabolites which are related to age, sex, BMI, and menopause as well as to participants’ diet.

## 1. Introduction

Metabolomics is as a powerful tool for phenotypic characterization of individuals, providing an unprecedented understanding of the molecular basis of human health. Over the last few years there has been a significant growth in the number of large-scale or epidemiological studies in the field of metabolomics aiming to better understand metabolic alterations at the population level [1,2]. Among the many challenges related to the design of an epidemiological-metabolomics study, one of vital importance is the assessment of the various sources of data variability that may interfere with the results and cause spurious or incorrect conclusions [1]. This is even more important for smaller-sized studies typically conducted in the clinical setting and, due to their size, are potentially more affected by an imbalanced or inappropriate study design. Over the last few years, many studies were thus conducted to investigate relationships between demographic characteristics, such as age, sex and body mass index (BMI), and blood metabolite concentrations, identifying the sets of metabolites being most affected by these covariates. These studies include several smaller-sized studies with less than 1000 participants [3,4,5,6,7,8,9,10], but also studies conducted in larger cohorts including Husermet [11] (age-related metabolites, 1200 individuals), WRAP [12] (age-related metabolites, 1212 individuals), KORA F3/F4 [13,14] (sex- and age-related metabolites, 2162 and 3300 individuals), KORA F4/SHIP [15] (sex-related metabolites, 1756 individuals), Framingham Offspring [16] (BMI-related metabolites, 2383 individuals) and Twins UK [13,17] (age-related metabolites, 6055 individuals). Some of the largest studies also investigated the relationship between metabolites and menopause status, including ALSPAC [18] (sex and menopause-related metabolites, 14,541 individuals), and the Northern European cohorts [19] (age-, sex-, menopause-related metabolites, 26,065 individuals).

Next to the aforementioned covariates, the blood metabolome can also be affected by environmental factors such as nutrition and diet. Studying associations between metabolites and dietary habits can thus also provide important information to be considered in the design and the analysis of metabolomics studies. As recently highlighted, nutritional metabolomics can help in the discovery of new biomarkers of nutritional exposure and status as well as in the comprehension of the molecular mechanisms by which diet affects the metabolome and consequently also health and disease [20]. Many nutritional metabolomics studies are designed as interventional, aiming to measure the specific output of a controlled diet in case x control/crossover studies [20]. At an epidemiological level, observational studies are more common. These studies usually focus on low versus high consumers of nutrients/foods assessed by food frequency questionnaires, food records, and other dietary assessment tools in order to discover biomarkers related to habitual food intake or to specific nutrients and food groups [21]. It is therefore of general interest to the metabolomics community to further understand which metabolites are mostly influenced by diet and, more specifically, by certain types of foods.

The Cooperative Health Research in South Tyrol (CHRIS) study [22] is a single-site population-based study with a longitudinal lookout aimed to investigate the genetic and molecular basis of age-related common chronic conditions and their interaction with lifestyle and environment, mainly focusing on cardiovascular, metabolic, neurological, and psychiatric health. The study is performed in the general population of a valley in South Tyrol (Italy) that is located in the middle of an Alpine mountainous area. The study population is characterized by some noteworthy features such as a homogeneous lifestyle and environmental conditions and low-residential mobility across generations.

In the present study, we present the Biocrates-based targeted metabolomics data set from the CHRIS study consisting of quantified metabolite concentrations for 175 metabolites in serum samples of 6872 study participants. We evaluated associations of metabolite concentrations with participants’ age, sex, and BMI in this large homogeneous single-site cohort of mostly healthy adult participants. We also identified metabolites related to the menopause status on female study participants. We found a considerable agreement between our results and the results reported by previous large-scale studies. At last, we evaluated relationships between diet and metabolite concentrations in a data subset for which food frequency questionnaire data was available.

## 2. Results

### 2.1. Study Sample Characteristics and General Data Overview

We created a targeted metabolomics data-set for 6872 participants of the CHRIS study [22] consisting of the quantification of 175 metabolites measured on 88 plates (see Appendix A for a listing of all metabolites including their names, aliases, chemical formulas and Human Metabolome Database (HMDB, https://hmdb.ca/, accessed on 1 February 2022) identifiers and Table 1 for the demographic characteristics of the study participants). To reduce plate differences, we normalized individual metabolite concentrations based on values measured in different quality control (QC) samples in the same batch, which reduced between-batch differences considerably, especially for the data acquired with flow injection analysis (see Appendix A). Normalization was able to reduce the average coefficient of variation (CV) across QC samples which were not being used for normalization from 9.8% to 9.2% (liquid chromatography-mass spectrometry (LC-MS) data) and from 25.8% to 10.9% (flow injection analysis (FIA) data) while not affecting the CV of study samples. After normalization, distribution of abundances was log normal for most analytes (see Appendix A for individual analytes’ signal distributions). Correlation between metabolite abundances showed the expected grouping by metabolite class (Appendix A). Clusters of metabolites with higher correlations can be spotted such as for branched chain amino acids (BCAA; leucine, isoleucine and valine) or for acyl-alkyl or diacyl phosphatidylcholines. We next evaluated the variability in measured metabolite concentrations in the present data-set. To this end, we calculated for each metabolite the ratio between the CV in study samples and the CV in QC samples hence accounting also for the technical variability of individual metabolites’ measurements. The largest difference between biological and technical variability was observed for biogenic amines ADMA, SDMA, DOPA, histamine and serotonin as well as carnitines C2 and C10 (see Appendix A). All of them, except histamine, had a CV in study samples higher than 35% while their technical variance was below 10%. In comparison, the median CV across metabolites in study samples was 27.8% and the median CV in QC samples 6.8%. In contrast, for many glycerophospholipids the biological variance was not much larger than their relatively high technical variance. Other metabolites, all with a technical variance below 10%, showed also a relatively low biological variance (below 20%). Among these were the acylcarnitines C12-DC, C10:2, C16-OH and amino acids His, Gln, Phe, Asn, Ser, Arg, Met, Lys and Val.

### 2.2. Sex, Age and Body Mass Index-Related Metabolites

We next aimed at identifying sex- age and BMI-related metabolites in our study sample. To this end we performed a multiple regression analysis fitting linear models explaining metabolite abundances by sex, age, fasting status and BMI category. We identified 42 metabolites with a significant difference in abundances between the 3747 female and 3125 male study participants (see Figure 1A and Appendix A). Metabolites with higher concentrations in females were predominantly from lipid classes while biogenic amines such as creatinine, amino acids (especially the branched chain Ile, Leu and Val) and acylcarnitines had higher concentrations in males. Except for citrulline and serotonin, all these metabolites were already identified to be sex-related in previous population-based metabolomics studies (see Appendix A). The analysis was repeated for the metabolite sums and ratios which summarize related metabolites or reflect important enzymatic reactions. Three out of the 10 sums and 8 out of the 23 ratios were found significantly related to sex (see Appendix A). Next, we determined the age-related metabolites in our data set. From the 175 metabolites, 148 were significantly associated with the participants’ age (Appendix A). Interestingly, almost all of them were positively associated with age, while for only 16 metabolites concentrations decreased with age (Figure 1B). Out of the 148 metabolites, 59 were already found to be age-related in previous studies (see Appendix A). Furthermore, all metabolite sums and ratios were found to be age-related (Appendix A). Finally, regarding BMI-related metabolites, we could identify 1, 7 and 29 with significantly different concentrations between BMI category 1 (underweight; BMI < 18.5, *n* = 99; Appendix A in the supplementary document), BMI category 3 (overweight; 25 ≤ BMI < 30, *n* = 2351; Appendix A) and BMI category 4 (obese; BMI ≥ 30, *n* = 1087; Appendix A) always compared against BMI category 2 (normal; 18.5 ≤ BMI ≤ 25, *n* = 3274). The difference in concentrations of these metabolites consistently increased (or decreased) with BMI (Figure 1C). From the in total 29 metabolites with significant differences in concentrations between any BMI group 16 were found in a previous large scale metabolomics study (see Appendix A). Regarding metabolite sums and ratios, 2 sums were found to be significant only in the comparison between BMI categories 4 and 2 while 2, 1 and 4 ratios were significant between BMI categories 1, 3 and 4 always compared against BMI category 2 (Appendix A).

### 2.3. Menopause-Related Metabolites

We next analyzed metabolite concentrations of female study participants to identify metabolites with significant differences in concentrations between pre- and postmenopausal women. We found 6 metabolites (Appendix A) all with significantly higher concentrations in postmenopausal women. However, 3 of them (Glu, Asp and C3) were also found significantly associated with a higher BMI and all of them were among the significant age-related metabolites. Because of the apparent relationship between age, BMI, and menopause, we repeated the analysis restricting to women in the menopausal transition period. We defined this period based on the age distribution of pre- and postmenopausal women in the present study sample (Figure 2) and included only women aged 49 to 57 into the analysis, which was thus based on 342 pre- and 312 postmenopausal women. In this analysis, 3 metabolites (C2, Glu and sarcosine) were found to have consistently higher concentrations in post-menopausal women (Appendix A). None of the significant metabolites were identified in a previous large-scale analysis [19], which can, however, be explained by the fact that the employed assay in that study did not allow quantification of these metabolites. In addition to the significant metabolites, also one metabolite sum (sum of acylcarnitines) and 4 metabolite ratios were found significant (Appendix A).

### 2.4. Metabolites Related to Food Items and Food Groups

Consumption frequencies of the different food items are presented in Appendix A. Foods and beverages with the highest consumption frequencies (servings per week > 10) included bread, vegetable oils, most fruits and vegetables, dairy products, coffee, and tea. In contrast, meat products, eggs, margarine, potatoes, and chocolate were consumed to a lower extent (servings per week ≤ 10). To identify diet-related metabolites we performed linear multiple regression analyses for all 8225 possible pairwise combinations of metabolites and food frequency questionnaire (FFQ)-derived food items and groups, respectively. From all tested combinations, 52 had a significant difference in metabolite concentrations between the highest and the lowest quintile of individuals with consumption of the food item (see Appendix A). These represent a total of combinations of 27 out of the 175 metabolites and 15 out of the in total 47 tested food items (Figure 3). Interestingly, most of the significant metabolites were lipids (19 phosphatidylcholines, 2 sphingomyelins and 3 lysophosphatidylcholines) and only 2 were amino acids and one a biogenic amine. For none of the tested carnitines was a significant relationship with high consumption of a food item found. Moreover, interestingly, certain patterns of relationships between metabolites and food items were apparent: a set of metabolites, mostly phosphatidylcholines, showed a positive relationship with higher red meat consumption, while 3 other phosphatidylcholines had a strong positive relationship with higher consumption of fish dishes. In contrast, another set of phosphatidylcholines showed a negative association with a higher consumption of poultry. Furthermore, a different set of lipids showed a positive relationship with higher butter consumption but a negative coefficient for margarine intake. Moreover, there was a strong positive relationship between serotonin concentration and consumption of herbal tea. Sensitivity analyses with further adjustments for BMI and seasonality only marginally affected our findings (data not shown).

## 3. Discussion

We created and analyzed one of the largest targeted metabolomics data-sets for a single-site cohort with 175 quantified metabolites in 6872 study participants. Employing a normalization strategy that used multiple quality control measurements per plate, we were able to reduce between-plate differences considerably, hence resulting in a final data set with a very low technical variability. Normalization specifically reduced the variability in FIA-based measurements. While enabling a faster and higher throughput analysis, FIA-MS/MS, compared to LC-MS/MS, suffers from higher matrix effects such as ion suppression, especially for more complex samples, hence resulting in a higher technical variability. Signal distributions were log-normal for most metabolites, even if measurements were partially outside of the quantification limits of the employed assay. Biological variability was above 35% for some biogenic amines and carnitines, suggesting a high inter-individual variability of these metabolites. Many amino acids, on the other hand, showed low biological variability in the present data set (below 20% with technical variability being below 10%). We also included the estimates for technical variance into the definition of significant metabolites for categorical variables in our analyses. For a metabolite to be called significant, we required, in addition to the statistical significance, that also the average difference in concentrations between the compared groups was at least twice as large as the (independently determined) technical variance for that metabolite in the same data set. We believe this additional criterion helps to avoid spurious findings and allows a focus on metabolites with larger and more reliable differences in concentrations. Overall, it was possible to observe a substantial overlap between our findings and the already-described metabolites, therefore reconfirming previously reported results on an independent cohort. In addition, novel metabolites were found for age, sex, and BMI, contributing not only to our understanding on how these metabolites behave in the study sample but also in possible future validation cohorts. Finally, the results obtained for our age-restricted analysis on pre- and post-menopausal women have revealed two novel metabolites associated with this transition period. It should however also be noted that the present analyses are limited to the set of metabolites quantified by the employed kit and information on certain important key metabolites might be missing in specific sub-analyses, such as for example steroid hormones in the menopause analysis.

Concentrations of individual metabolites might also be influenced by lifestyle, medication, or by the presence of certain diseases. The number of participants with severe health conditions is however expected to be very low in our study sample. Still, we cannot rule out that some of the identified associations might also be affected by these additional factors. This might specifically be true for age or BMI-related metabolites, since general health is expected to decrease with these. The decline in health is generally also paralleled with an increase of medications for related diseases. The most common medications in our data set consist of agents acting on the renin-angiotensin system, antithrombotic agents, or thyroid therapy (the average age of participants for these medications ranges from 56–69 years; see Appendix A for a full listing). About 10% of the participants take one of these medications, but with different frequencies or dosages. Furthermore, these represent only high-level therapeutic categories, while the actual prescribed drugs within these categories might act on different metabolic pathways, making it hard to accurately assess their influence on the measured metabolite concentrations.

### 3.1. Sex, Age and Body Mass Index-Related Metabolites

The results for sex, age and BMI-related metabolites are summarized in Figure 4. A large proportion of the studied metabolites showed significantly increasing concentrations with age. A considerable number of metabolites was also found to be shared between the different studied covariates, which is not unexpected given that these variables are to some extent also related to each other.

Regarding sex-related metabolites, the sum of branched chain amino acids (tBCAA, incudes Ile, Leu and Val) showed a strong difference in the concentrations between males and females. Furthermore, the three BCAA presented the lowest *p*-values across all amino acids (AAs) when comparing individual metabolite concentrations. These AAs were previously described to have higher concentrations in male subjects [23]. Similarly, creatinine also had a significant, and on average 23% higher, concentration in males. It is well known that creatinine levels are affected by many factors, including not only sex but also age, diet, medication, and muscle mass [24]. We also compared the concentration of this metabolite with the values obtained by clinically certified quantitative tests which were available for CHRIS study participants and found a high correlation between these (R = 0.849). This further shows the high quality of the present data-set.

Focusing on metabolites/metabolite ratios found to be higher in females subjects, two were worth highlighting: Serotonin/Trp and DOPA/Tyr. The Serotonin/Trp ratio, representing the activity of the tryptophan hydroxylase, was found to be, on average, 24% higher in female subjects, with Trp levels being significantly lower in females, while serotonin concentrations were significantly higher. In addition, the DOPA/Tyr ratio, representing the activity of the tyrosinase enzyme, was found to be 9% higher in female participants, due to the significantly lower Tyr levels found in male participants (DOPA concentrations were about the same in both groups). Both DOPA and serotonin are classified as neurotransmitters and have been previously associated with neurological disorders, such as depression and anxiety [25,26]. Moreover, the need for considering sex-related factors in future anxiety studies has been highlighted [27]. While there are still contradictory results in terms of the availability and metabolism of these metabolites [27], it has been shown that sex hormones such as estrogen, testosterone and progesterone play a role in the serotonergic system (and most likely on the DOPA/Tyr system). Finally, serotonin has also been associated with both lean mass (positive correlation) and fat mass (negative association) in males. The same study found no significant association in females, unless they are restricted to post-menopausal women, which can also be an indication of the influence of estrogen on serotonin levels. The conclusion by the authors is that there are sex differences in how serotonin interplays with body fat distribution [28]. These results are in line with our sex-related results on serotonin, and also with the finding that serotonin was negatively associated with BMI in our data. In addition to significantly higher concentrations in female participants, serotonin was also one of the few metabolites with a negative relationship with age. More precisely, we found for serotonin an average decrease of plasma concentration of 0.87% per year. Many studies have already described the age-related decrease of serotonin and its receptors [29] but, to the best of our knowledge, we report here for the first time quantitatively the average decrease of serotonin concentrations in plasma with age. Moreover, Trp was also found to be negatively related with age, with an estimated concentration decrease of 0.23% per year.

Moreover, some results on metabolite ratios are worth highlighting: the Fischer ratio (tBCAA/tAromAA, where tAromAA includes Phe and Tyr) was found to be significantly lower in female participants. This was expected since, tBCAA was found to be, on average, 20% higher in males. A lower Fischer ratio was previously described to be indicative of a higher liver dysfunction [30] and a higher incidence and severity of adverse cardiac events in patients with heart failure [31]. Our findings should, however, not be interpreted as an indication for a higher risk of these pathologies’ incidences in female participants, but rather as an indicator that the Fisher ratio should be considered separately for male and female individuals. Similarly, the Orn/Arg ratio was found to be on average 17% higher in males, driven mostly by Orn, since Arg concentrations were about the same in both sexes. Considering that Orn is mainly produced from Arg via the arginase enzyme, this result might suggest a higher activity of this enzyme in males. As summarized by Caldwell [32], arginase hyperactivity was previously associated with many different outcomes, including hypertrophied and fibrotic hearts and kidneys, vascular dysfunction, pulmonary hypertension, neural toxicity and cancer growth. More recently, Goita and coauthors have described the sexual dimorphism of metabolite in patients with arterial hypertension, also highlighting the arginine/NO pathway [33]. In addition to these findings, we have found many metabolites from the Orn/Arg pathway to be also age-related. Cit, SDMA and Orn were all among the most significant metabolites related to age. Moreover, the ratio Cit/Arg had the lowest *p*-value across all the analyzed ratios and Orn/Arg, the 3rd lowest *p*-value. All these metabolites and metabolite ratios were found to be increasing with age and they are all common to the urea cycle, which seems to be of great interest for many diseases/phenotypes. Figure 5 summarizes our findings on metabolites in this pathway. Other metabolites closely related to this cycle which were also found to be significantly related to age and/or sex in our analysis are Pro, which was significant for sex (23% higher in males), the ratios Putrescine/Orn and Spermidine/Putrescine significant for age, with a decrease of 0.52% and 0.17%/year, respectively and spermidine significantly decreasing with age by 0.12% per year.

Lastly, BMI-related metabolites were identified by comparing average metabolite concentrations of participants from the BMI categories 1 (underweight), 3 (overweight) and 4 (obese) to category 2 (normal). For the first comparison between underweight and normal, only Tyr was found to be significant. This metabolite also showed increasing concentrations from categories 1 to 4 (see Figure 1C). More specifically, compared to category 2, average concentrations were by 9% lower in category 1, and higher by 7% and 17% in categories 3 and 4 (effect sizes −0.41, 0.32 and 0.72, respectively). Dunn and co-authors [11] have previously reported Tyr levels to be significantly different when comparing between BMI < 25 and BMI > 30 groups (with n ≈ 1200). The authors also pointed out that Tyr, together with Phe and Val (which have also been found to be significant in their comparison), are potential early markers of insulin resistance and risk for the development of diabetes. These two metabolites were also significant between obese and normal BMI categories in our data-set. Furthermore, it was interesting to notice that all 7 metabolites significant for overweight participants were also among the 29 metabolites significant for the obese group. Unique metabolites listed for category 4 include sugars, Ile, Leu (among many other amino acids), which have all been found at higher levels in obese participants and have also been previously associated with insulin resistance and a risk factor for diabetes [34]. Overall, we observed increasing metabolic dysregulation with increasing BMI, both in the number of significant metabolites as well as the extent of the difference in concentrations.

### 3.2. Menopause-Related Metabolites

The menopause period is characterized by a multitude of metabolic changes in women’s bodies, led mainly by the reduction of estrogen and progesterone levels [35,36]. These have previously been associated with higher Glu levels in women during the menstrual cycle [37]. We also found concentrations of Glu to be significantly higher in postmenopausal women supporting this finding. In contrast, the two other significant metabolites, C2 and sarcosine, have, to our knowledge, never been associated with menopausal status so far. For C2 it is, however, known that it can up-regulate the expression of the metabotropic glutamate receptor 2 (mGlu2), which reduces glutamate release from primary afferent sensory fibers, resulting in an analgesic effect. For this reason, C2 is being used in the treatment of neuropathic pains [38]. These two metabolites seem thus to be linked to some extent to menopausal status and neurological pathologies [39,40].

The third significant metabolite was sarcosine, also showing higher concentrations in postmenopausal women. Sarcosine is primarily synthetized from choline, via betaine and dimethylglycine intermediates inside the choline oxidation pathway [41]. In turn, choline levels are mainly diet-dependent, since only a small portion can be endogenously synthetized via de novo biosynthesis of phosphatidylcholine, catalyzed by the phosphatidylethanolamine-N-methyltransferase (PEMT) enzyme. The activity of this enzyme was shown to be sensitive to estrogen levels [42] and, consequently, premenopausal women are more resistant to choline deficiency than postmenopausal women [43]. Our sarcosine results suggest a potential link between menopause and the choline oxidation pathway. While our dataset does not include measurements of betaine and dimethylglycine, Gly, the metabolic product of sarcosine, was marginally significant (the adjusted *p*-value was below 0.05, but the difference in concentrations was just below the cut-off) with a higher average concentration in post-menopausal women. This smaller coefficient of glycine, compared to the one of sarcosine, may be explained with the fact that most endogenous glycine is produced via serine and not sarcosine.

Summarizing, our results suggest a menopausal dysregulation of both energy and neuro-associated metabolism. Further studies are required to elucidate the potential role of these metabolites in menopause-related changes of metabolism and its relationship to neurological conditions.

### 3.3. Metabolites Related to Food Items and Food Groups

Most of the food items found to be significantly associated with a metabolite were related to meat intake (beef, fish, pork, poultry, and processed meat). Among the metabolites found to be significantly associated with higher consumption of pork, beef, and processed meat was trans-4-hydroxiproline, an amino acid that is highly abundant in beef and negligible in plant-source foods [44]. Indeed, trans-4-hydroxyproline was already previously described as a potential biomarker for red meat intake in the Twins UK cohort [45].

In addition, we found higher intake of pork and processed meat to be negatively associated with glycine, which plays an important role in metabolic regulation, anti-oxidative reactions, and neurological functioning [46]. Similarly, low serum glycine levels were also found to be associated with total red meat consumption in the EPIC Potsdam cohort [47]. Moreover, in our study red meat intake was positively associated with two unsaturated acyl-alkyl phosphatidylcholines (PC), namely PC ae C36:5 and PC ae C36:4. In contrast, higher poultry consumption was negatively associated with another set of PC’s (PC ae C36:1, PC ae C36.2) and LysoPC C17:0. Meat is a rich source of phosphatidylcholine [48], which can be converted by the intestinal microbiota into trimethylamine and subsequently into trimethylamine oxide-N-oxide (TMAO). Previous studies have identified TMAO to be a potential biomarker for meat intake, meat-containing diets, as well as fish intake [5].

Regarding fish consumption, associations were more compound than class specific. Both PC aa C34:3 and PC aa C40:5 were found to be inversely related to fish intake, while PC aa C38:0, PC aa C38:6 and PC aa 40:6 showed strongly positive coefficients for that food item. The latter 3 metabolites were previously reported to be positively correlated to a dietary pattern characterized by a high intake of fish and poultry and a low intake of sweet foods, margarine, tea, and whole-grain bread [49]. Similarly, PC aa C38:6 and PC aa C40:6 were found to be positively correlated with oily fish consumption [49]. Finally, Altmaier and co-authors [50] have found, across the analysis of the same set of PCs, the ratio PC aa C40:5/PC aa C40:6 to be the strongest (negative) association with fish intake.

High butter consumption was significantly related to yet another set of metabolites (LysoPC a C17:0, three acyl-alkyl phosphatidylcholines and 2 sphingomyelins). Interestingly, margarine consumption was negatively associated with these metabolites (Figure 3), one of them, namely LysoPC a C17:0, was also significant. This metabolite was also previously described to be positively associated with consumption of cream [45] and with a dietary pattern characterized by high butter and high-fat dairy intake and low margarine consumption. LysoPC a C17:0, being mainly derived from milk fat, has thus also previously been suggested to be a biomarker for dairy fat intake [49], which is in line with our findings.

Regarding plant-based foods, we found a higher intake of pome fruits (pear, apple), potatoes and bread to be negatively associated with a set of diacyl Phosphatidylcholines (PC aa C38:4/C36:5/C36:6). PC aa C36:6 was also among a set of glycerophospholipids previously described to be associated, although positively, with a dietary pattern characterized by a higher intake of fruits and vegetables in women [17]. No other significant associations were found with other plant foods in our study. Recently, also using the GA^2^LEN FFQ, a metabolomic study of 17 adults identified several serum markers of dietary intake of sources of fruits and vegetables in adults, including rhamnazin 3-rutinoside, 2-galloyl-1,4-galactarolactone methyl ester, 2″,32″-di-O-p-coumaroylafzelin and cyclocommunin. Additionally, the bio-compound 2-galloyl-1,4-galactarolactone methyl ester was strongly correlated with total flavonoid intake. Pending further replication in larger study samples, these novel potential biomarkers might be used in population-based surveys as markers of dietary intake of flavonoids, fruits and vegetables [51]. Other associations between metabolite and plant-based foods have also been mainly reported based on small scale intervention and cross-sectional studies, such as *S*-methyl-l-cysteine sulfoxide or proline betaine as biomarkers of cruciferous vegetables or citrus fruit [21]. It should however be noted that the biospecimen used in these studies was mainly urine (after fasting). Moreover, some of these biomarkers might represent short-term intakes rather than long-term dietary habits [20].

When considering beverages, we found a rather unexpected, positive relationship between serotonin and higher green and herbal tea consumption. To exclude this relationship being potentially caused by seasonality, assuming a higher tea consumption in winter as well as a potential season-dependency of serotonin concentrations, we re-evaluated this relationship adjusting also for the season of participation, but the associated *p*-value and coefficient were only marginally affected. Further studies are required to investigate how plasma serotonin levels are influenced by herbal or green tea consumption. Among other associations with beverages were the positive relationships between both beer and wine consumption and several PC aa’s, some of which (PC aa C32:1 and PC aa C36:5) were also previously reported to be related to wine consumption [45]. In addition, we found a negative association between wine and glycine, and beer and lysoPC a C17:0.

Summarizing, we observed a relatively strong relationship between habitual meat intake and metabolite concentrations, despite lower consumption frequencies ranging from an average of 2.9 to 9.8 servings per week. This might in part be explained by a rather stable meat consumption over longer time periods. Our findings are in line with previous studies, suggesting trans-4-hydroxyproline to be a good candidate biomarker for habitual red meat intake. Another interesting finding was that margarine with a very low consumption frequency (2.7 servings per week) had such a strong effect on the metabolite LysoPC C17:0, whereas consumption of other vegetable oils with a much higher reported consumption frequency (21 servings per week) had no such effect. Interestingly, most fruits, vegetables and grain products with higher consumption frequencies did not show any significant association with any metabolite. Exceptions from plant-based foods and grains were found for pome fruits, potatoes, and bread. We might assume the consumption of these latter three food groups to be more regular throughout the year as the availability of these food items is not strongly restricted to seasonality as compared to other fruits and vegetables. Indeed, we observed that fruits like citrus fruits were most affected by seasonality (Appendix A). This also shows that responses to food frequency questionnaires, although they might be expected to represent annual averages, can be biased by the season of participation. Participants tend to overestimate consumption of food items more frequently consumed at the time they compile the FFQ, suggesting that, at least for some food groups, the data might need to be adjusted by season. In general, although comparisons of food-related results across studies is difficult because of non-standardized questionnaires, our nutrition-related findings are in line with previous studies investigating food intake and metabolites.

## 4. Materials and Methods

### 4.1. Participant Recruitment and Sample Collection

CHRIS participants were recruited from the adult population of the middle and upper Vinschgau/Val Venosta district located in the northern-most region of Italy, close to the border with Switzerland and Austria. Sample collection and self-reported, questionnaire-based, health assessment was performed in a single recruitment center in the middle of the study area. The barcode of the medication boxes study participants brought along were scanned and the respective Anatomical Therapeutic Chemical (ATC) codes were assigned. Thus, medication information classified by the hierarchical ATC coding system was collected for all participants. Collected blood samples were aliquoted and serum and plasma samples were obtained for posterior biobanking. Details on the population, the recruitment process, sample handling and phenotyping/health assessment are provided in Pattaro et al. [22]. Between 2011 and 2018, 13,393 participants were enrolled in the study.

Dietary intake over the past 12 months was assessed for a subset of CHRIS study participants with the Global Allergy and Asthma European Network (GA^2^LEN) study food frequency questionnaire (FFQ) [52]. The FFQ is organized into 32 sections covering 195 specific food items and beverages. Participants were asked to self-report how often, on average, they consume each food item indicated in the FFQ, given standard portion sizes following the Food Standard Agency Food Portion Sizes Guidelines [53], with response options ranging from: rarely or never; 1–3 times per month; once per week; 2–4 times per week; 5–6 times per week; once per day; and 2+ times per day. For our analysis on food items, consumption frequencies were converted to servings per week. We further grouped food items into 47 food groups (Appendix A) based on the similarity of foods. Total energy intake (kcal/day) was calculated by linking the FFQ to the latest edition of the Composition of Foods Integrated Dataset [54]. All individuals with >20% missing FFQ items or with a total energy intake to basal metabolic rate ratio <0.5th or >99.5th centile were excluded from this specific analysis.

Prior to the analyses, food consumption expressed in servings/week was adjusted for total baseline energy intake using the nutrient residual method [55].

### 4.2. Sample Preparation and Data Acquisition

Serum samples were prepared using the AbsoluteIDQ^®^ p180 kit from Biocrates (Biocrates Life Sciences AG, Innsbruck, Austria) following the manufacturer’s protocols. Briefly, 10 µL of serum samples, quality control samples (both provided by Biocrates and in-house prepared by pooling all analyzed samples after collection step) and calibration curve samples were transferred to Biocrates filter plate using Tecan Fluent 780 automatic pipetting system (Tecan Trading AG, Männedorf, Switzerland). One double blank sample (Milli Q water) and one blank sample (PBS) were also added. Samples were further handled according to a protocol provided by Biocrates, which includes an initial filtering step followed by a derivatization employing phenyl isothiocyanate and the sample extraction with 5 mM ammonium acetate solution in MeOH. Subsequently, sample aliquots are transferred into two distinct 96-well plates and filled with LC-MS and FIA-specific dilution solutions. Samples were analyzed with a UHPLC-MS/MS system consisting of an Agilent 1290 Infinity LC system (Agilent Technologies, CA, USA) and a Sciex QTRAP 6500 mass spectrometer (AB Sciex LLC, MA, USA). For LC-MS analysis, mobile phases constituted of 0.2% formic acid in water (mobile phase A) and 0.2% formic acid in acetonitrile (mobile phase B). The flow rate for the whole analysis was set to 0.8 mL/min (unless otherwise specified) and the following gradient elution (%A) was used: 100% (0 to 0.45 min), from 100 to 85% (0.45 to 3.30 min), from 85% to 30% (3.30 to 5.90 min), from 30% to 0% (5.90 to 6.05 min), 0% (6.05 to 6.20 min), 0% (6.20 to 6.42) with an increase in the flow rate to 0.9 mL/min, 0% (6.42 to 6.52) with a decrease in the flow rate to 0.8 ml/min, 0% to 100% (6.52 to 6.70 min) and 100% from 6.52 to 7.30 min). For the FIA-MS/MS analysis, isocratic elution was employed using a mixture of methanol and Biocrates solvent from glass ampoule. Flow rate was changed over the analysis as follows: 200 µL/min (0 to 0.09 min), from 200 to 35 µL/min (0.09 to 0.1 min), 35 µL/min (0.1 to 0.9 min), from 35 to 150 µL/min (0.9 to 1.40 min), from 150 to 1000 µL/min (1.40 to 1.45 min), 1000 µL/min (1.45 to 1.88 min), from 1000 to 200 µL/min (1.88 to 1.91 min). Sciex Analyst 1.7 software bundle was used for instrument control and data collection. Each sample was analyzed with absolute quantification of 42 metabolites (21 amino acids and 21 biogenic amines) by LC-MS/MS and semi-quantification of 146 metabolites with FIA-MS/MS (40 acylcarnitines, 90 glycerophospholipids, 15 sphingolipids and 1 sum of hexoses). Note that for glycerophospholipids and sphingolipids each reported “analyte” represents in fact the sum of all possible isobars and structural isomers. Abbreviations represented as C x:y indicates the number of carbons (x) and the number of double bonds (y). Moreover, glycerophospholipids are characterized according to the presence of an ester (a) or ether (e) bond in the glycerol moiety. Finally, double letters (aa = diacyl, ae = acyl–alkyl) indicate that two glycerol positions are bound to a fatty acid residue, while a single letter (a = acyl or e = alkyl) indicates a bond with only one fatty acid residue. For UHPLC-MS/MS data, integration of peaks was visually verified using the Sciex Multiquant software and results were imported together with FIA-MS/MS data into the Biocrates MetIDQ software. Final raw concentrations were exported for additional data quality assessment and normalization.

### 4.3. Data Preprocessing

Each 96 well plate contained measurements of metabolites in 80 study and 4 different sets of quality control (QC) samples: Biocrates p180_QC1 and p180_QC2, an aliquot of the SRM 1950 (Metabolites in Frozen Human Plasma) from NIST (QC NIST STD) as well as an aliquot of the pool of all serum samples of the study (QC CHRIS Pool). A normalization factor for each metabolite on each plate was calculated based on the concentration of this metabolite in all QC samples, except QC NIST STD (which was used as an independent measurement to evaluate normalization performance). In detail, a normalization factor was first calculated for each QC sample type separately based on the mean concentration of the 3–5 measurements that were available for this QC sample type on a single plate. These were then averaged across the 3 employed QC sample types (using the mean) to result in the final normalization factor for the plate and metabolite. For quality assessment, the coefficient of variation (CV) in QC samples, the number of missing values as well as the distribution of the signal per plate and in the whole data-set was considered. Thirteen of the 188 metabolites were excluded because of poor quality, reducing the data set to 175 metabolites. In addition to the individual metabolite measurements, 10 metabolite sums and 23 ratios representing summaries of related metabolites or reflecting important enzymatic reactions were calculated based on the sum and ratios definitions from Biocrates (Appendix A). For subsequent data analysis, all reported concentrations were used, even if they were outside of the quantification limits of the Biocrates assay. Missing metabolite concentrations because of a signal below the detection limit of the instrument were replaced with a random number from a uniform distribution ranging from the smallest measured value for that metabolite to half of this value. The final metabolomics data-set, after excluding data from pregnant women (*n* = 34), consisted of concentrations for 175 metabolites in 6872 individuals.

### 4.4. Statistical Analysis

All analyses were performed on log2-transformed metabolite concentrations. To identify sex, age and BMI-related metabolites, linear regression models were fitted separately for each metabolite using its concentration as response variable and sex, age, fasting status and BMI as covariates. BMI was included as a categorical variable with the four categories underweight (category 1, BMI < 18.5, *n* = 99), normal range (category 2, 18.5 ≤ BMI < 25, *n* = 3274), overweight (category 3, 25 ≤ BMI < 30, *n* = 2351) and obese (category 4, BMI > 30, *n* = 1087); for 61 participants no BMI information was available. Category 2 (normal range) was used as reference level. Self-reported fasting information was used for the binary fasting status variable, with 6415 participants declaring to not have had a meal in the 12 h prior blood drawing while 455 did (2 participants with missing data). While storage time can have an influence on concentrations of certain metabolites we opted to not adjust for this factor because we assumed the storage conditions to be unrelated to the investigated traits of interest (e.g., assuming independence between storage time and age or sex of participants). To identify metabolites related to menopausal status we performed linear regression analyses on the data subset of female participants (*n* = 3747). Women either declaring to be in the menopause or that were aged older than 56 and had ceased menstruation due to an operation were classified as being postmenopausal (*n* = 1267). The linear models included the metabolite concentration as response variable and menopause status, age, fasting status and BMI (categorical) as explanatory variables. To reduce the influence of the apparent relationship between age, BMI and menopause, an additional analysis was performed focusing only on women in the menopause transition phase. Based on the age distribution of pre- and postmenopausal women (Figure 2), we defined the menopausal transition period for the current study to be between 49 and 57 years of age and performed a linear regression analysis on this data sub-set consisting of 342 pre- and 312 postmenopausal women employing the same model as above but without age as a covariate.

For the relationship between metabolites and diet, food intake (expressed as servings per week) was categorized into quintiles, with quintile 1 (representing the 20% of the study participants with the lowest reported food intake) as the baseline. Linear regression models were applied with the metabolite concentration as response variable and categorized food intake, age, sex, and fasting status as explanatory variables. In our main analysis, BMI was not included in the set of covariates as it could rather be considered a mediator instead of a confounder, lying on the causal pathway between dietary intake and metabolite levels. However, to assess the robustness of our main models, we conducted a sensitivity analysis by further adjusting the models for BMI. In addition, seasonality, could affect both metabolites and dietary intake. We therefore further investigated whether food intake responses from the FFQ were influenced by seasonality. To this end, logistic regression models were fitted separately for each food item using its binarized concentration (with class “0” and “1” for food intake values being below or above the median for the specific food item) as response variable and sex, age, fasting status, BMI and seasonality as covariates.

*p*-values from all analyses were adjusted for multiple testing using the Bonferroni correction method. Metabolites with an adjusted *p*-value smaller than 0.05 were considered significant. For categorical variables (sex, BMI, fasting status and food item consumption), a difference in abundances between categories (expressed in percentage) was required, which was at least twice as large as the coefficient of variation in QC samples for the given metabolite. The coefficient of variation was calculated on the QC CHRIS Pool samples, hence representing the technical variability that was observed for a metabolite in the present data-set.

### 4.5. Comparison of Significant Metabolites with Results from Literature

Biocrates identifiers were manually mapped to Human Metabolome Database (HMDB) IDs considering the compounds’ names, chemical formulas and additional information extracted from Biocrates’ technical documentation. Metabolites found significant in previous large cohort-based studies [11,12,13,14,15,16,17,18,19] were collected and annotated to Biocrates identifiers using their name (either directly if Biocrates IDs were provided or by matching the name to the compounds’ aliases) or based on their HMDB ID (if provided). To ensure that results for individual metabolites across studies represent the same direction of relationship than presented here, the sign of coefficients was adapted depending on the reference level that was used in the statistical model. Literature results, additional annotations and summary statistics for the metabolites analyzed in this study are provided in Appendix A.

## 5. Conclusions

We presented here one of the largest single-site targeted metabolomics data-sets with 175 quantified metabolites in 6872 participants of the CHRIS study. We could reconfirm most of the findings from previous large scale metabolomics studies on the relationship of metabolite concentrations on age, sex, BMI, menopause, and FFQ-based food consumption. Such reconfirmation and replication of scientific findings in an independent cohort is pivotal for their validation and ultimate consolidation. Over and above, the high-quality metabolomic data-set presented here might prove to be a valuable resource for following studies as well as for large-scale consortia. A portal with the aim of enabling a structured and simplified way to request access to data from the CHRIS study, including the complete metabolomics data as well as the genotype data, is currently being implemented.

## Figures and Tables

**Figure 1 metabolites-12-00205-f001:**
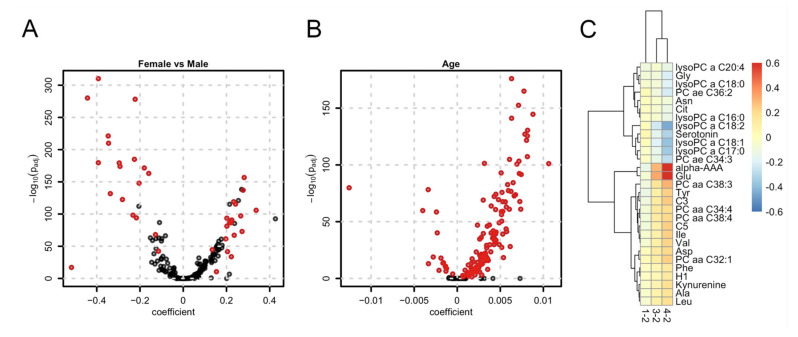
Sex-, age- and BMI-related metabolites. (**A**) Volcano plot for the differential abundance of metabolites between females and male participants. The coefficient represents the log2-difference in average concentrations between the groups; (**B**) Volcano plot for age dependency of metabolites. The coefficient represents the log2-change of concentration per year; (**C**) Heatmap of coefficients for metabolites found to be significant for at least one BMI category. The coefficient represents the average log2-difference in abundance between BMI categories 1, 3, and 4 compared to the reference category 2.

**Figure 2 metabolites-12-00205-f002:**
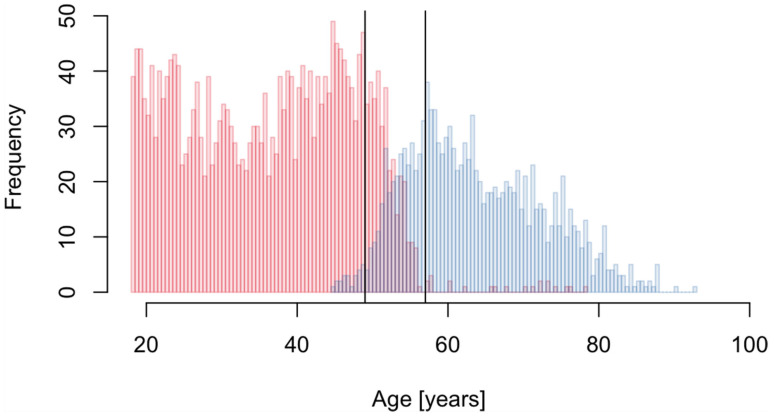
Age distribution of pre- (red) and postmenopausal (blue) women. The vertical solid lines indicate the selected menopause transition period (49 to 57 year).

**Figure 3 metabolites-12-00205-f003:**
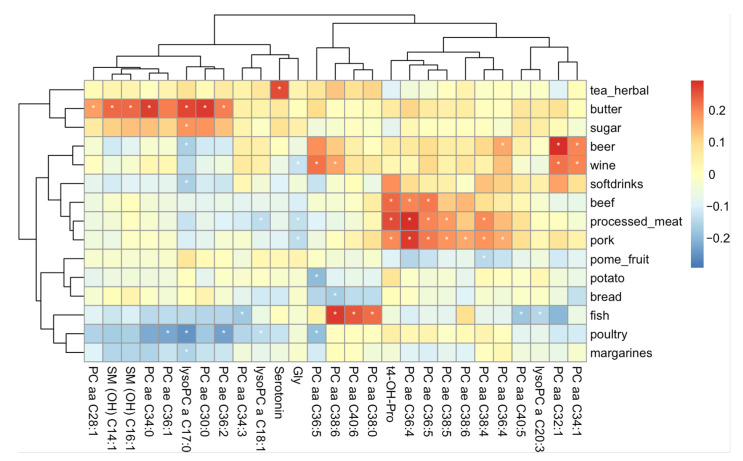
Heatmap of coefficients representing the difference in metabolite concentrations between the top and bottom 20% of individuals with the highest and lowest consumption of a food item. Only food items (rows) and metabolites (columns) are shown with at least one significant relationship. Significant coefficients are indicated with a white asterisk. Red coloring represents higher metabolite concentrations in participants with a higher consumption of the food items, blue coloring the opposite.

**Figure 4 metabolites-12-00205-f004:**
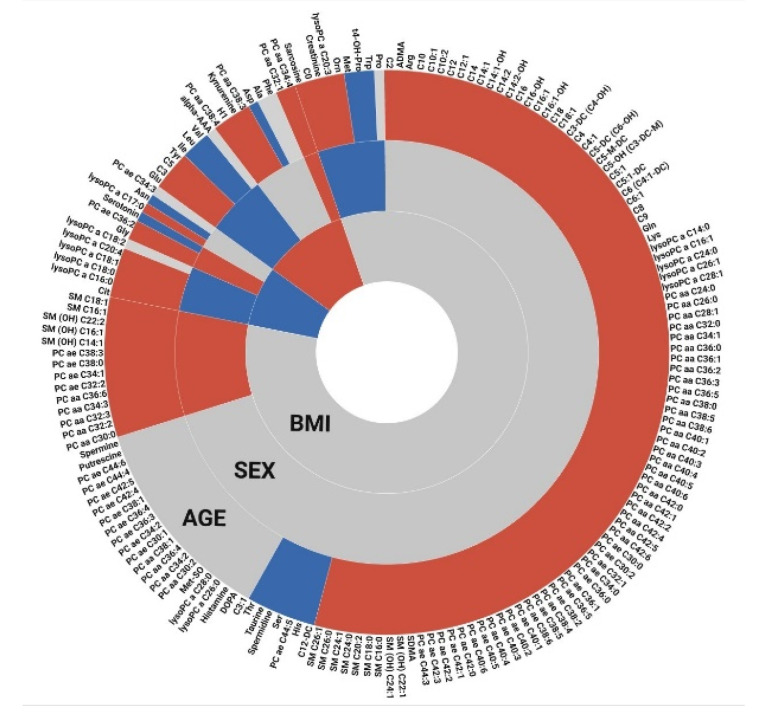
Overview on metabolites related to age, sex, and BMI. Gray spaces represent non-significant metabolites, while blue and red colors represent negative and positive coefficients, respectively. For sex-related metabolites, female is considered the baseline, i.e., metabolites with higher concentrations in male participants are colored in red (and vice versa). For BMI-related metabolites, the results from the comparison of obese against normal are shown.

**Figure 5 metabolites-12-00205-f005:**
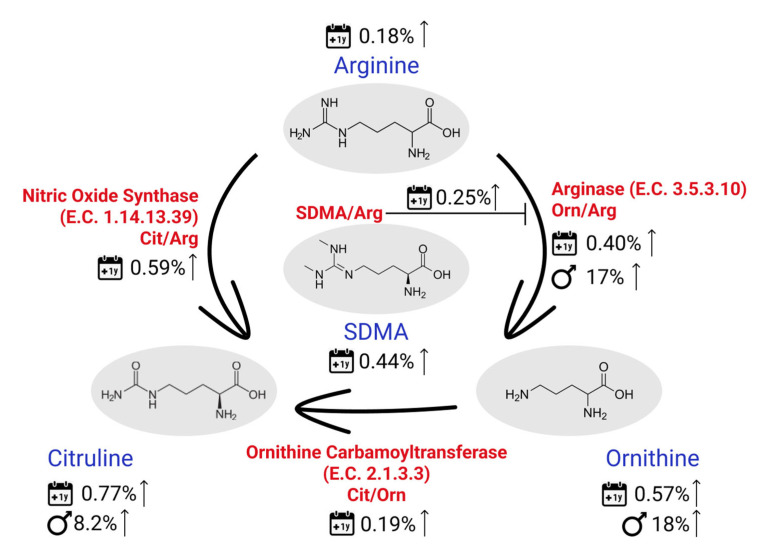
Sex- and age-related (1-year span) findings on the main metabolites and metabolites ratios involved in the urea cycle.

**Table 1 metabolites-12-00205-t001:** Demographic characteristics of the study participants included in the analysis.

	Metabolomics Data		Metabolomics and Diet Data	
	Male	Female	Male	Female
*n*	3125	3747	1095	1231
Age [mean (SD)]	46.4 (16.5)	45.7 (16.5)	46.4 (16.7)	45.0 (16.6)
Not fasting, *n* (%)	222 (7.1%)	233 (7.4%)	98 (8.9%)	70 (5.7%)
BMI, *n* (%)				
1: underweight	13 (0.42%)	86 (2.3%)	5 (0.46%)	32 (2.6%)
2: normal	1202 (38.5%)	2072 (55.3%)	421 (38.5%)	660 (53.6%)
3: overweight	1374 (43.9%)	977 (26.0%)	487 (44.5%)	335 (27.2%)
4: obese	503 (16.1%)	584 (15.6%)	180 (16.4%)	202 (16.4%)
missing	33	28	2	2

## Data Availability

The data presented in this study are available upon request to the CHRIS Access Committee. Contact the corresponding author for details. Full results of the present analyses are provided in the Appendix A.

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
