# Peer review of "Age, Sex, Body Mass Index, Diet and Menopause Related Metabolites in a Large Homogeneous Alpine Cohort"

_metabolites, 2022, doi:10.3390/metabo12030205_

Round 1

Reviewer 1 Report

Concentrations of metabolites are influenced by age, sex, menopause, diet and many other factors. A better understanding of these relationships is important for study design and data analysis. The authors identified metabolites associated with age, sex, body mass index, diet, and menopausal status using 175 quantified metabolites in 6,872 study participants. This work confirmed findings from previous population-based studies and reported some novel associations which may improve our current understanding of metabolic phenotypic changes associated with healthy aging.

Comments:

  1. Batch correction of data from large-sized samples is technically challenging, but crucial to subsequent analysis. In the inter-batch RLA plots (Fig S1), it seems that the LCMS was more stable than FIA. What were the possible reasons for the “suboptimal” performance of FIA? Have the authors conducted any pretreatment of the pooled QCs before using them for batch normalization?
  2. Why BMI category but not the BMI value was used in the linear regression models to identify sex-, age- and BMI-related metabolites? Additionally, it is better to add the storage time of biological samples to all the models as this may be a significant factor to the concentrations of some metabolites measured, especially for a long-term sample collection process.
  3. A total of 175 metabolites, 10 classes, and 23 ratios based on definitions from Biocrates were involved in this study. Have the authors considered additional variables such as pathway activities and more metabolite ratios to be generated respectively according to KEGG pathways and using the Lilikoi R package (http://3mcor.cn/utools.html). Based on the extended variable set, more differential variables with rich biological implications will be observed.
  4. The study population was from a valley in South Tyrol (Italy) with a homogeneous lifestyle and environmental conditions and thus the findings were area-specific. It is of special interest to the metabolomics community to understand the impact of special lifestyle and environment to your findings, especially the newly identified associations and the inconsistent results with existing reports. For example, in previous reports (ref 12, the WRAP cohort), the numbers of metabolites associated to age and gender were comparable while in this study, there was many more age-related metabolites compared to gender.
  5. Please provide more descriptions about the health assessment and medications or med histories of participants which may affect data homogeneity.
  6. This report provided a comprehensive literature summary of metabolites (with plenty of additional information) associated to age, gender, BMI, and diet. An open and web-based database may provide the research community with a better access to the resource.

Reviewer 2 Report

The article is devoted to an important aspect of intraspecific changes in the serum concentration of a number of important metabolites. This study is characterized by a large sample, powerful statistical data processing and detailed study of the results. Despite this, there is little actual novelty in this work, which the authors themselves mentioned several times in the article. The problem of weak novelty is associated with the presence of a number of previously conducted similar studies, which the authors mention in the introduction and when comparing the data obtained. Also a limitation in terms of novelty is the determination of 175 metabolites using a commercial kit. However, these limitations are also the strengths of this study. Metabolic profiling, performed by looking for unknown compounds in chromatograms obtained after chromatography-mass spectrometric analyzes, often raises questions, since pre-validation of the method is usually not even provided. Thus, the validity of the observed differences is usually inconclusive. The presented study is characterized by the reliability of the obtained chromatographic data and subsequent patterns, although it is limited to a set of 175 metabolites.  Of course, it is interesting to learn about differences in the concentration of other classes of compounds, such as polyphenols, fatty acids and other compounds associated with diet and microbiota characteristics in people of different sex, age and body mass index.  Also, an important advantage of the work is the study of the obtained material, the identification of biochemical patterns and analysis of the literature and HMDB. It is necessary to add a list of abbreviations used in the article, as well as a transcript in the text at the place of mention. In general, the manuscript leaves a good impression of the volume and quality of the study. 

Reviewer 3 Report

In this manuscript, Hernandes et al. profiled the serum metabolome of >6000 individuals. The authors discovered several interesting biomarkers, which are related with age, sex, BMI and menopause. This is a large-sized metabolomics study. The presentation quality of this manuscript is good. The statistical analysis is also well conducted. I just have some comments listed below:

  • The coverage of metabolites is still limited. Although amino acids and biogenic amines are very important metabolites, but other class of metabolites (e.g. short chain fatty acids, carbohydrate, carboxylic acids, lipids) are also important. For example, as authors mentioned in the manuscript, menopause may be strongly related with steroid hormones, which are not covered in this metabolomics study. The authors may consider to expand the coverage of metabolome in the future study.
  • The authors should provide details of the LC-MS settings. (e.g. gradient, mobile phase).
